

# Assessment of the potential of drug-drug interactions among population-based oldest-old people in Turkiye

Fuat Nihat Özaydın[1] and Ayşe Nilüfer Özaydın[2]

[1] Pharmacology Department, Istanbul Atlas University Faculty of Medicine, Istanbul, Turkiye
[2] Public Health Department, Marmara University Faculty of Medicine, Istanbul, Turkiye

## ABSTRACT

**Background:** The risk of potential drug–drug interactions is highest in oldest-old people. Thus, the aim of this study was to investigate the frequency and type of potential drug–drug interactions in population-based oldest-old people.

**Methods:** The type of study was descriptive. Ethical permission was obtained (13.04.2022/153). All participants were informed, and their written consent was obtained. The "oldest-old" were defined as those who were ≥85 years of age during the study period and living in Turkiye. These people were reached from every region of Turkiye *via* the snowball method and were visited at their homes. Data were collected *via* face-to-face interviews. Age, sex, city of residence, and generic names of regularly used medications were recorded. The medications used were analyzed according to the Beers 2019® Criteria and UpToDate® Lexicomp® drug interaction guides. SPSS was used for statistical analysis, and $p < 0.05$ was considered statistically significant.

**Results:** Data were collected from the 549 oldest-old people throughout Turkiye. Among the participants, 61.2% ($n = 336$) were women. The median age of the women was 88.00 years (minimum = 85, maximum = 101), and the median age of the men was 88.00 years (minimum = 85, maximum = 102). The distributions of men and women in the different age groups were similar ($p = 0.341$). The distributions of men and women across regions were similar ($p = 0.826$), most of whom ($n = 300, 54.6\%$) had ≥1 potential drug–drug interaction, according to the UpToDate analysis. The median number of medications used continuously was 4.0 (minimum = 0, maximum = 19). The median number of potential drug–drug interactions was 1.0 (minimum = 0, maximum = 21). As the number of medications used increased, the number of potential drug–drug interactions also increased ($r = 0.737; p = 0.001$). The number of potential drug–drug interactions decreased with increasing age ($r = -0.104; p = 0.015$). According to the Beers 2019® Criteria, potential drug–drug interactions were detected in only eight patients. The concordance between the Beers 2019® Criteria and the UpToDate®Lexicomb® drug interaction data was poor compared with the number of potential drug–drug interactions (kappa = 0.024, $p < 0.001$). Central nervous system medications are a common group that can cause potential drug–drug interactions according to both guidelines. Moreover, potentially inappropriate medications defined by the Beers 2019® Criteria were the most common causes of potential drug–drug interactions, according to UpToDate®Lexicomb® drug interactions. The frequency of potential drug–drug interactions was found to be high in the population-based oldest-old people

Corresponding author
Fuat Nihat Özaydın,
fnozaydin@gmail.com

interviewed in Turkiye. It has been determined that the use of more than one guide in the evaluation of potential drug–drug interactions is safer.

## INTRODUCTION

The populations of Turkiye and the world are aging (*World Health Organization, 2024*; *Turkish Statistical Institute (TURKSTAT), 2021*). The oldest-old group will be the group that will grow the fastest within the aging population (*World Health Organization, 2024*), and this population is expected to increase at least threefold within 30 years (*World Health Organization, 2024*; *Eurostat, 2020*). The American Geriatrics Society and the World Health Organization accept the oldest-old as ≥80 years of age. The British Geriatrics Society and the European Statistical Office define the oldest-old individuals as ≥85 years of age (*Escourrou et al., 2022*; *Eurostat, 2020*). The Turkish Statistical Institute defines the oldest-old as those ≥85 years of age (*Turkish Statistical Institute (TURKSTAT), 2021*). The oldest-old group deserves great attention among older people, as this is the most vulnerable group in need of health services (*Sang Bum et al., 2018*).

Cognitive impairment or functional disability was found in half of the individuals in the oldest-old group. Nutritional status was abnormal for one quarter of this population, and physical status was abnormal for one-third to one-half of the participants in the oldest-old group (*Escourrou et al., 2020*). The multimorbidity that facilitates the development of these symptoms increases with age. The proportions of men and women with ≥2 chronic diseases were 47.4% and 46.9% in the <75-year age group and 70.0% and 73.3% in the 75–84-year age group, respectively. This percentage increased to 76.3% and 88.1% in the ≥85-year age group, respectively (*Kirchberger et al., 2012*). Polypharmacy is emerging in the treatment of multimorbidity. In one study, the mean numbers (±SD) of diseases and prescribed medications in the 65–79-year age group were 5.5 ± 2.1 and 6.5 ± 1.9, whereas they were 14.2 ± 9.9 and 17.6 ± 10.4, respectively, in the ≥80-year age group (*Mo et al., 2016*). In another study examining the 85+ year group, the numbers of diseases and prescribed drugs were found to be 6.4 and 6.8, respectively (*Tsoi et al., 2014*).

The multimorbidity rate was 62% in the 65–74-year age group, increasing to 81.5% in the ≥85-year age group in Turkiye. The most common health problems in this age group are hypertension, diabetes, coronary artery disease, chronic pain, chronic kidney disease, and depression. Different medications, such as antipsychotics, antidepressants, hypnotics, analgesics, and laxatives, are frequently prescribed for multimorbidity treatment (*Turkiye Elderly Health Report: Current Status, Problems and Short-Medium Term Solutions, 2021*). The analysis of prescriptions registered with the Ministry of Health of Turkiye revealed that 16.1% of the ≥85-year age group received ≥4 prescriptions (≥5 medications) over 1 year, suggesting the existence of chronic polypharmacy in oldest-old people (*Aydos et al., 2020*).

There is a risk of potential drug–drug interactions (pDDIs) due to polypharmacy in the oldest-old population. While the risk of DDIs is 13% with the combination of two drugs, it increases to 38% with the combination of four drugs and to 82% with the combination of ≥7 drugs (*Lavan & Gallagher, 2016*). The risk of pDDIs increases with increasing age, and the risk is highest in the oldest-old period. In one study, the risk of DDIs was 24% in the 60–79-year age group and increased to 36% in the ≥80-year age group. Another study analyzing 330,000 older people (≥65) reported that the risk of pDDIs increased with increasing age. While the odds ratio for severe pDDIs was 1 for the 65–69-year age group, it was 1.74 for the ≥85-year age group. Similar to these studies, the odds ratio for DDI exposure was 1.07 (95% CI [1.03–1.11]) in the 70–74-year age group and increased to 1.52 (95% CI [1.46–1.60]) in the ≥85-year age group in one study (*Anrys et al., 2021*; *Guidoni et al., 2012*; *Nobili et al., 2009*).

Medications can be purchased from pharmacies without a prescription and used for self-treatment in Turkiye. It was reported that 41% of medications were purchased from pharmacies without a prescription in Istanbul (*Gul et al., 2007*). Medications purchased without a prescription or reused are not recorded in the person's health records (*Yuksel, Ozaydin & Ozaydin, 2022*). For these reasons, an observational study involving population-based people should be performed to analyze the medications used (*Cıbık, Sahin & Kılıncaslan, 2018*). In a study with population-based older people, it was determined that age ≥80 years was associated with ≥3 pDDIs (*Santos et al., 2017*).

Finally, there are studies explaining multimorbidity and polypharmacy in the oldest-old people throughout Turkiye. However, no study has examined pDDIs between the medications used by the oldest-old people through Turkiye. The purpose of this study was to examine the frequency and types of pDDIs due to medications used by the population-based oldest-old people and to analyze their relationships with the demographic characteristics of the participants.

## METHODS

### Ethics

This study used the datasets collected for our previous studies (*Özaydın, 2024a*, *2024b*). New ethical approval (13.04.2022/153) was received from the same ethics committee (Istanbul Okan University Ethics Committee) for the reuse of these datasets for the analysis of pDDIs. Written informed consent was obtained from all participants. The study was conducted in accordance with the principles of the Declaration of Helsinki.

### Study design, setting and population

Study design, setting, characteristics of interviewers, calculation of population size, selection of participants, demographic characteristics, geographic distribution of participants, recording of medications used and other information have been reported (*Özaydın, 2024a*, *2024b*).

## Instrument applied

In this study, the Beers 2019® Criteria and UpToDate® Lexicomp® drug interaction guides were used for pDDI analysis (*The 2019 American Geriatrics Society Beers Criteria® Update Expert Panel, 2019*; *Wolters Kluwer, 2022*). The Beers 2019® Criteria describe contraindicated pDDIs in older individuals. The UpToDate database rates pDDIs using a risk rating of A (unknown interaction, no data), B (no action needed, little to no evidence of clinical concern), C (monitor therapy, dosage adjustment of one or both agents), D (consideration to modify therapy, aggressive monitoring, empirical dosage changes, choosing alternative agents), or X (contraindicated, avoiding combination). According to the Beers 2019® Criteria, contraindicated pDDIs correspond to X based on the UpToDate classification.

In the elderly group, central nervous system drugs are frequently used together. The use of three or more psychotropic drugs, such as antidepressants, antipsychotics, hypnotics, sedatives, and anxiolytics, together occurs at a frequency of up to 39% (*Bergman et al., 2007*). The only guide that analyzes drug–drug interactions in the use of ≥3 central nervous system drugs is the Beers guide (*The 2019 American Geriatrics Society Beers Criteria® Update Expert Panel, 2019*). For this reason, the Beers 2019 guide was preferred. The UpToDate guide has been shown to be valid in studies conducted with older community-dwelling individuals (*Hughes et al., 2023*). Additionally, it has been shown to detect drug–drug interactions between prescription and nonprescription drugs in population-based studies (*Tayanny Margarida Menezes Almeida et al., 2022*). This approach was preferred in our study due to these findings.

## Statistical analysis

The data were analyzed *via* the IBM SPSS Statistics 29.0.0.0 software package (IBM Corp., Armonk, Chicago, IL, USA). For descriptive analyses, categorical data are expressed as $n$ (%). The Kolmogorov–Smirnov test was performed as a normality test, and nonparametric tests were used for non-normally distributed continuous data. Non-normally distributed data are presented as median, minimum, and maximum values. Mann–Whitney U tests, Kruskal–Wallis tests, Pearson's chi-square tests, and Fisher–Freeman–Halton exact tests were performed, and Spearman's correlation coefficient was calculated. A $p$ value of <0.05 was considered to indicate statistical significance. The concordance between the two criteria was calculated *via* the kappa test. Kappa values >0.75 were considered excellent, values <0.75–0.40 were considered fair, and values <0.40 were considered poor (*Ma et al., 2019*).

## RESULTS

Data were collected from 549 oldest-old people throughout Turkiye. There was participation from all geographical regions (five) of Turkiye. The median age of the participants was 88.00 years (minimum = 85, maximum = 102; $n$ = 549). Among the participants, 61.20% ($n$ = 336) were women. The distribution of men and women according to age groups and geographical regions was similar ($p_{\text{CHI-SQUARE, FISHER-FREEMAN-HALTON}}$ = 0.341, and $p_{\text{CHI-SQUARE, FISHER-FREEMAN-HALTON}}$ = 0.826, respectively).

**Table 1 Distribution of number of medications and pDDIs of the oldest-old people according to the UpToDate Classification.**

| Number of medications | Number of oldest-old people (%) |
|---|---|
| 0 | 15 (2.8) |
| 1 | 47 (8.5) |
| 2–4 | 281 (51.1) |
| 5–9 | 175 (31.9) |
| ≥10 | 31 (5.7) |

| Number of pDDIs | Number of oldest-old people (%) |
|---|---|
| 0 | 249 (45.4) |
| 1 | 115 (21.0) |
| 2–4 | 115 (21.0) |
| 5–9 | 55 (10.0) |
| ≥ 10 | 15 (2.6) |

| Number of medications | Minimum and maximum number of pDDIs |
|---|---|
| 2–4 | 0–5 |
| 5–9 | 0–11 |
| ≥10 | 3–21 |

## pDDIs, according to the UpToDate classification

Most of the oldest-old people (88.7%, $n = 487$) used ≥2 medications, and 54.6% ($n = 300$) of those had ≥1 pDDI, according to the UpToDate analysis (Table 1). The median number of medications used by participants was 4.0 (minimum = 0, maximum = 19), and the median number of pDDIs was 1.0 (minimum = 0, maximum = 21). The number of pDDIs increased as the number of medications used by participants increased ($r_{SPEARMAN} = 0.737$; $p = 0.001$).

The median number of pDDIs and the frequency of pDDIs among the oldest-old group did not change with sex ($p_{MANN\ WHITNEY\ U} = 0.612$ and $p_{CHI-SQUARE} = 0.481$, respectively) or region ($p_{KRUSKAL-WALLIS} = 0.953$ and $p_{CHI-SQUARE,\ FISHER-FREEMAN-HALTON} = 0.822$, respectively). The median number of pDDIs was 1 in the 85–89 and 90–94 age groups and 0 in the 95–99 and 100–104 age groups ($p_{KRUSKAL-WALLIS} = 0.077$). The frequency of pDDIs tended to decrease in the 100-year and above age groups compared with the younger age groups (56.1%, 53.7%, 47.5%, and 16.7% in the 85–89, 90–94, 95–99, and ≥100-year age groups, respectively) ($p_{CHI-SQUARE,\ FISHER-FREEMAN-HALTON} = 0.261$). Additionally, the total number of pDDIs among the oldest-old people decreased toward the ≥100-year age group ($r_{SPEARMAN} = -0.104$, $p = 0.015$) (Fig. 1).

The total number of pDDIs among the medications used by the 300 oldest-old people was 962. Most pDDIs were in Group C of the UpToDate pDDI classification (Table 2). The highest number of pDDIs observed in one person was 21. Twenty of these interactions belonged to Group C, and one belonged to Group X, according to the UpToDate pDDI classification. The second person had 3B+15C (18 total) and the third person had 3B+12C +2D (17 total) types of pDDIs according to the UpToDate classification. More than one

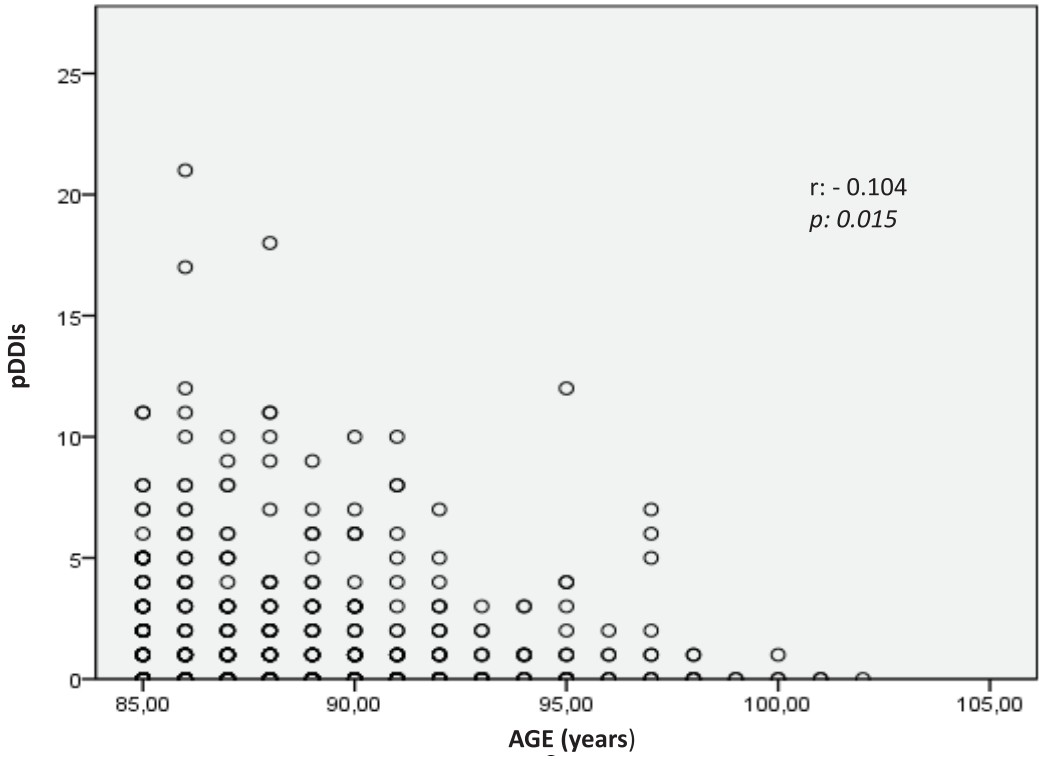

**Figure 1 The correlation of age and the number of pDDIs.**

**Table 2 Distribution of oldest-old people and pDDIs according to the type of pDDIs specified in the UpToDate classification.**

| Types of pDDIs | Number of oldest-old people (%) | Number of pDDIs (%) |
|---|---|---|
| C | 263 (47.9) | 730 (75.9) |
| B | 108 (19.6) | 135 (14.0) |
| D | 60 (10.9) | 74 (7.7) |
| X | 21 (3.8) | 21 (2.2) |
| A | 2 (0.3) | 2 (0.2) |
| All types of pDDIs | 300 (54.6)* | 962 (100.0) |

Note:
* More (the same and/or different groups) than one pDDI coexisted in one oldest-old person.

pDDI (in the same and/or different groups) coexisted in one oldest-old person. Table 3 shows the distribution of pDDIs among the oldest-old people.

According to the UpToDate classification, 5.1% of pDDIs were detected in association with antibiotics belonging to the fluoroquinolone group (moxifloxacin, ciprofloxacin, interaction with 20 different molecules) used for infectious diseases.

Other pDDIs are the result of molecules used for noncommunicable diseases. The top five pharmacological agents that caused pDDIs were antipsychotics (36.9% of the oldest-old people, interactions with 34 different molecules), beta-blockers (24%, interactions with 39 different molecules), thiazide-type diuretics (22.5%, interactions with 32 different

**Table 3 Distribution of the types of pDDIs in the UpToDate classification among the oldest-old people.**

| One pDDI | n (%) | Two pDDIs | n (%) | Three pDDIs | n (%) | Four pDDIs | n (%) |
|---|---|---|---|---|---|---|---|
| C | 146 (80.7) | B+C | 55 (63.3) | B+C+D | 18 (62.0) | B+C+D+X | 2 (66.7) |
| B | 22 (12.1) | C+D | 23 (26.5) | B+C+X | 8 (27.6) | A+B+C+D | 1 (33.3) |
| D | 12 (6.7) | C+X | 7 (8.0) | C+D+X | 2 (7.0) | | |
| X | 1 (0.5) | D+X | 1 (1.1) | A+B+C | 1 (3.4) | | |
| | | B+D | 1 (1.1) | | | | |

molecules), selective serotonin reuptake inhibitors (SSRIs) (22.4%, interactions with 38 different molecules), and nonsteroidal anti-inflammatory drugs (NSAIDs) (22%, interactions with 21 different molecules, and NSAID-NSAID combinations). Table 4 shows the most frequently found pDDIs, including their type and clinical effect, according to the UpToDate guidelines.

## pDDIs, according to the Beers 2019® Criteria

pDDIs were detected in only eight (1.4%) patients. Anticholinergic drug combinations (paroxetine+olanzapine, paroxetine+solifenacin, and cyproheptadine+chlorpheniramine) and ≥3 medication combinations, such as antidepressants, antipsychotics, and benzodiazepines, were types of pDDIs (0.5% and 0.9%, respectively). Finally, anticholinergics, antihistamines, and central nervous system drugs used as potentially inappropriate medications (PIMs) according to the Beers Criteria cause pDDIs.

## Comparison of the UpToDate classification and Beers 2019® Criteria

According to the Beers 2019® Criteria, contraindicated groups of pDDIs were found in only eight (1.4%) patients, whereas the X group of pDDIs (avoiding combination) was detected in 21 (3.8%) patients, according to the UpToDate classification.

Two of the anticholinergic-type interactions, which were in the contraindicated group according to the Beers 2019® Criteria, were listed in Group C (monitor therapy) according to the UpToDate classification. However, the interaction of paroxetine and solifenacin identified by the Beers 2019® Criteria as a contraindicated pDDI was not included in the UpToDate classification. Even if three or more central nervous system drugs, such as antidepressants, antipsychotics, and benzodiazepines, are used together, the UptoDate guidelines only analyze the interaction as a dual combination. The interaction was classified as C. The Beers Criteria were used to analyze the interaction of the ≥3 drugs and were classified as contraindicated. Both guidelines showed that central nervous system medications frequently cause pDDIs. Furthermore, PIMs (fluoroquinolones, antipsychotics, SSRIs, thiazide-type diuretics, and NSAIDs) defined by the Beers 2019® Criteria were the most common causes of pDDIs, according to UpToDate. The kappa statistics for the Beers 2019® Criteria and UpToDate classification indicated poor coherence compared with the number of pDDIs (kappa = 0.024, $p < 0.001$).

**Table 4 The most frequently reported pDDIs, including their type and clinical effect, according to the UpToDate guidelines.**

| Drug-drug interaction | n | Type of pDDIs | Possible clinical effect |
|---|---|---|---|
| Antipsychotics (Quetiapine, olanzapine, risperidone)–acetylcholinesterase inhibitors (rivastigmine, donepezil) | 25 | C | Decreased efficacy of acetylcholinesterase inhibitors used to treat dementia, increased in the case of severe extrapyramidal symptoms |
| Aspirin low dose-ACE inhibitors (ramipril, perindopril, lisinopril, zofenopril, cilazapril, trandolapril) | 25 | C | Decreased renal function, decreased antihypertensive effect of ACEIs, increased nephrotoxicity effect |
| Nonsteroid anti-inflammatory drugs (NSAIDs: ibuprofen, meloxicam, tenoxicam, nimesulide, aceclofenac, etodolac, diclofenac, flurbiprofen, naproxen, dexketoprofen)-ACE inhibitors (ACEIs)/Angiotensin 2 receptor blockers (ARBs) (ramipril, perindopril, enalapril, captopril, lisinopril, valsartan, olmesartan, candesartan, losartan, irbesartan) | 23 | C | Decreased renal function, decreased antihypertensive effect of ACEIs, decreased antihypertensive effect of ARBs |
| Metformin-Beta 1 blockers (propranolol, carvedilol, metoprolol, nebivolol, bisoprolol) | 18 | C | Increased risk of hypoglycemia, masking of hypoglycemia symptoms, reduced antidiabetic therapeutic effect |
| Antipsychotics (quetiapine, olanzapine)-selective serotonin reuptake inhibitors (escitalopram, citalopram, sertraline) | 17 | C | QT prolongation, increased risk of neuroleptic malignant syndrome, increased risk of serotonin syndrome |

## DISCUSSION

This was the first nationwide study in which data were collected from a cohort of population-based oldest-old people in real-life conditions. Based on our literature research, this was also one of the pioneering studies showing the high risk of pDDIs for central nervous system medications and PIMs. Moreover, a "healthy survival effect" was found in the advanced oldest-old group.

A study that included 4,222,165 people living in the Emilia-Ramagna region of Italy analyzed medications registered by the Italian National Health Service in 2004 (*Gagne, Maio & Rabinowitz, 2008*). The pDDI list prepared from the extant literature was used for the analysis, and 8,894 pDDIs were identified in a cohort of 7,902 individuals. The percentage of pDDIs among the 957 oldest-old people (≥85 years) was 69%, similar to our result (54.6%). pDDIs were observed in more than half of the participants. In both studies, fluoroquinolone antibiotics were the medication group associated with pDDIs. Fluoroquinolones are included in the PIM group for older people. Ciprofloxacin, a member of the fluoroquinolone group, has been included in the PIM group in the Beers Criteria since 2019 due to the risk of drug–drug interactions (*The 2019 American Geriatrics Society Beers Criteria® Update Expert Panel, 2019*). According to the EU-7 PIM list, fluoroquinolone antibiotics are included in the "Questionable PIMs" list (*Renom-Guiteras, Meyer & Thürmann, 2015*).

A study published in 2015 analyzed community prescriptions for 310,000 adults written by general practitioners. These prescriptions were registered with the National Health Service between 1995 and 2010 (*Guthrie et al., 2015*). The British National Formulary was used as a reference, and the frequency of potential serious interactions (avoided or undertaken only with caution and with appropriate monitoring) was analyzed. In this

study, 17,977 of the participants were in the oldest-old group (80+ years of age). Potentially serious pDDIs were detected in 46.0% of this group. In our study, 51.1% of the oldest-old people had pDDIs in categories C, D, or X- as similar categories (monitor therapy, consideration to modify therapy, or contraindicated, respectively). The results of the two studies were similar.

A study published in 2012 in Norway investigated the prevalence of pDDIs by examining 8,268 elderly people. The average age of these elderly individuals was 83.0 years, and they were receiving home care services. A Norwegian web-based tool was used for the analysis. DDIs were detected in 57% of the participants, and DDIs of the "should not be combined" group were detected in 2%, which was similar to the results of our study. In our study, 54.6% of the participants had all groups of pDDIs, and 3.8% of the participants had the X group (contraindicated) (*Halvorsen et al., 2012*).

A study of 181 outpatients with dementia reported similar results. The mean age of the participants was 80.11 years. The Micromedex Drug Reax 2.0® database was used. The molecules that frequently cause pDDIs are antidementias, antipsychotics, and antidepressant drugs (*Bogetti-Salazar et al., 2016*). These molecules were similar to those identified in our study. The frequency of drug–drug interactions was also similar (54.6% *vs.* 59.1%, respectively).

A general physician performed a home medication review of 779 home-dwelling elderly people during a home visit in Germany. The **A**rbeitsgemeinschaft der **B**erufsvertretungen **D**eutscher **A**potheker (ABDA) database was used for pDDI analysis. With the >75–80 year age group set as the reference, the moderate or severe pDDI odds ratio was 1.016 in the >80–85 year age group and 1.077 in the >85–90 year age group but decreased to 0.378 in the >90 year age group (*Hoffmann et al., 2011*).

Another study analyzed serious (should be avoided) pDDIs of the medications used by 732,228 older people (≥75 years). These older groups were registered in the Swedish Prescribed Drug Register system between October and December 2005. When the 75–79-year age group was used as the reference, the pDDI odds ratio was 0.92 in the 80–84 year age group, 0.86 in the 85–89 year age group, and 0.78 in the ≥90 year age group. Advanced age is associated with fewer pDDIs than younger age (*Johnell et al., 2007*). This finding has been termed the "healthy survival effect" (*Johnell & Fastbom, 2008*). Our study results supported this finding. The frequency of pDDIs among the oldest-old people tended to decrease with increasing age, and the total number of pDDIs observed in those individuals also decreased with increasing age. One reason for the "healthy survival effect" may be that the advanced oldest-old people with high-quality medication therapy or a reduced need for treatment can reach the centenarian age group (*Johnell & Fastbom, 2008*). Population-based studies reporting that the use of medications and PIMs (including pDDIs) decreases with age after 85 years also support this finding (*Grina & Briedis, 2017*; *Blozik et al., 2013*).

As in the comparative studies, the medications causing pDDIs in our study were mostly used to treat noncommunicable diseases. The rate of multimorbidity due to chronic diseases in elderly individuals (≥65) is 46% in Turkiye. Due to multimorbidity, older people visit many different specialists with short examination durations. A short visit time

increases the risk of polypharmacy and DDIs (*Turkiye Elderly Health Report: Current Status, Problems and Short-Medium Term Solutions, 2021*). Prescriptions written in primary health care institutions and registered with the Ministry of Health in 2018 were analyzed. The rate of chronic polypharmacy (prescriptions written ≥4 per year with ≥5 drugs) in the oldest-old (≥85) group was 16.1% (*Aydos et al., 2020*). All these data from Turkiye support the high pDDI rates reported in our study.

There are several important differences and similarities between the two guides used for pDDI analysis in the present study. For example, the UpToDate classification considered 2-drug combinations, contrary to the Beers 2019® Criteria, which considered 2-drug combinations and ≥3 central nervous system drug combinations for the detection of pDDIs. Using two guides together will provide a more appropriate analysis of daily life. Polypharmacy and ≥2 central nervous system drugs prescribed together are common in the oldest-old people.

An important difference was in the combined use of two NSAIDs. Although the NSAID +NSAID combination was included in the X pDDI group according to the UpToDate classification, it was not included in the Beers Criteria. Furthermore, the Beers Criteria include the combination of paroxetine and solifenacin, which are not included as pDDI in UpToDate. When the combination of paroxetine and solifenacin was evaluated according to the **A**nti **C**holinergic **B**urden (ACB) calculator, it was recommended that this combination be changed due to the high risk of confusion, falling, and death, depending on the anticholinergic effect (*ACB Calculator, 2023*). These findings highlight the poor concordance between the UpToDate classification and the Beers 2019® Criteria for the determination of pDDIs and support the use of more than one guide in the analysis of pDDIs.

The study was conducted nationwide and included information on age, sex, geographical distribution, and all medications used. This study is the first national study to show that central nervous system medications and PIMs carry a high risk of causing pDDIs. There was a "healthy survival effect" among the centenarians in this age group. These findings are the strengths of our study.

The interviewed sample may not be representative of all the oldest-old people living in Turkiye because the data were collected *via* the snowball sampling technique, which is a nonprobability sampling method. Not all the oldest-old individuals in the Turkiye had an equal chance of being selected. The study was conducted between December 2021 and May 2022, and the medications used by participants may be affected by seasonal effects. No prospective follow-up was performed in terms of adverse effects or hospitalization. The physician writes the medications on a computer system, and the pharmacist gives those to the patient using the system. Therefore, the patient did not have any proof such as a prescription in our study. Moreover, the oldest-old people were not asked whether the medications were prescribed due to the possibility of forgetfulness, misremembering, and low health literacy. Therefore, the classification of medications as prescribed or not was not made. These limitations should be considered when interpreting the study results.

## CONCLUSION

Polypharmacy is common in the oldest-old people in Turkiye. The frequency of pDDIs was also high. When pDDIs are prescribed for older people, including the oldest-old group, pDDI analysis must be performed *via* more than one guide.

### Funding

The authors received no funding for this work.

### Competing Interests

The authors declare that they have no competing interests.

### Author Contributions

- Fuat Nihat Özaydın conceived and designed the experiments, performed the experiments, prepared figures and/or tables, authored or reviewed drafts of the article, and approved the final draft.
- Ayşe Nilüfer Özaydın analyzed the data, prepared figures and/or tables, authored or reviewed drafts of the article, and approved the final draft.

### Ethics

The following information was supplied relating to ethical approvals (*i.e.*, approving body and any reference numbers):

The University of Istanbul Okan granted Ethical approval to carry out the study within its facilities (Ethical Application Ref: 13.04.2022/153).

### Data Availability

The raw data are available in the Supplemental Files.

### Supplemental Information

Supplemental information for this article can be found online at http://dx.doi.org/10.7717/peerj.19032#supplemental-information.

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
