# Peer review of "Assessment of the potential of drug-drug interactions among population-based oldest-old people in Turkiye"

_PeerJ, doi:10.7717/peerj.19032_

## Round 0.1 · original submission · Major Revisions

There reviewers have identified a number of concerns that need to be addressed. To my mind the most important of these are around the ethics approval, which appears to post-date much of the data collection, and the quality of the data collection - as reviewer 1 points out, there were a large number of people collecting what is not actually a great deal of data, and it is important to understand whether this lead to heterogeneity in the data collected.

The authors should also clarify whether they collected information about the medications that participants bought for themselves rather than having them prescribed. This is rightly identified as a potential confounder when creating an accurate picture of potential medication interactions from health records alone; my assumption is that this information was collected, can the authors confirm this ? If it was, including an analysis of what potential interactions arise from prescribed medication versus what might arise from self-sourced medications could be a valuable addition to the work.

The authors should address all the recommendations of the reviewers, and in particular remember that the interactions identified in the study are potential ones only, and no data was collected to indicate whether anything of consequence was actually occurring.

·

Basic reporting

The introduction section provides background about the oldest-old population, the prevalence of medication use, and drug-drug interactions. However, information on the need for this study is missing. The authors need to connect the existing information from the already published literature with the need for the study to set an adequate background for the readers. They should explain the purpose of conducting this study and link it with the study objectives.

Additionally, in lines 71-75, the authors highlight the unaccounted/unrecorded self-medication practices in Turkey and the need for the observational study. However, in the study limitations section, the authors, highlight the same issue as one of the limitations of this study (line nos. 226-227), which indicates that the authors were unable to fill the gap (that they identified earlier) through this study.

Experimental design

After reading this manuscript several times I felt that the authors attempted to asses the potential drug-drug interactions as a sub-analysis in the main study that assessed the potential inappropriate medication use among the oldest-old population. As the study results did not include any information about the outcomes of the drug-drug interactions in the studied population, it will be more appropriate to update the study title to "Assessment of potential drug-drug interactions (pDDIs) among population-based oldest-old people in Turkey". Additionally, the authors must use the word "potential drug-drug interactions (pDDIs)" throughout the manuscript instead of drug-drug interactions (DDIs).

Line no. 82 provides details on the ethical approval of this study. The approval number 13.04.2022/153 is different than the approval number for the main study as mentioned in the reference numbers 15 (Approval No: 141, Date: September 8, 2021) & 16 (08.09.2021/141). However, the rest of the methodology remains the same and the authors have also redirected the readers to these literatures for more details on the methodology of this particular study (lines no. 87-95). The authors need to clarify the mismatch and the reasons for the same.

I feel that the study data is part of a large study that assessed the potentially inappropriate medication use among the oldest-old population of Turkey, which the authors also mentioned in the methods part (references 15 & 16). But what is required is, that the authors clearly define the objectives of this particular study, then related methods involved and results. This means that the authors need to maintain a flow of thoughts that are required for this study's objectives rather than making the reader to refer different literature related to this publication (ref. no. 15 & 16). For example, line nos. 94-95, mentions that "Moreover, the frequency of anticholinergic medication, and potentially inappropriate medication use were also examined in detail", which is I feel out of scope for the contents of this particular manuscript, as this manuscript does not contain any results/data related to this objective. Therefore, the authors must limit the objectives and methodology involved in this study, in the patients and methods section.

References 15 & 16, also mentioned that the study data was collected across Turkey by the 120 students as interviewers. The authors need to mention and justify, what was the quality control and quality assurance method/s that they adopted to ensure that the data collected by these 120 interviewers was accurate, adequate, and quality data.

Lines nos, 98-102, mention the study tools used. The authors have used Beers criteria in the main study to assess the potentially inappropriate medication use among the study population. Table 4. of Beer's criteria provides details on contraindicated drugs for older patients because of the increased risk of drug-drug interactions (DDIs) but does not mention the different levels and severities of DDIs. On the other hand, UpToDate provides detailed information on interacting drugs, severity, and management strategies as well. With this background, the authors need to explain the rationale for comparing the concordance among these two resources. The concordance between two resources can be performed when both are for the same purpose, with reasonably similar depth of information. The authors may mention/compare the observations of pDDIs as per UpToDate and the missing or additional information that they have found in the Beers' criteria making some drug combinations as contraindicated, but concordance can not be explored.

Lines 107-110 mention the statistical test involved. The authors need to mention and explain which statistical analysis was carried out for which type of data in detail for the benefit of the readers. The authors also need to elaborate on the procedure they adopted to explore the concordance between Beers' criteria and UpToDate, for example, which parameters were considered for evaluating the concordance and how it was done.

Validity of the findings

Line nos. 112-113, directs the readers to other related publications for the basic details of the study participants, which I feel inappropriate. The readers must have all the necessary and adequate information on the required results of this study in this manuscript only. The authors need to present all such details in the current paper rather than redirecting the readers elsewhere.

Line nos. 114-115, "The median age of the participants in the study was 88.00 (range= 85-102; n=549)". The number 88 must be followed by "years" and the range must be mentioned as "interquartile range" or abbreviated as "IQR". This change must be done wherever authors have mentioned "range" throughout the manuscript.

Line nos. 119-120, "The number of DDIs increased as the number of medications used by participants increased (r=0.737; p=0.001)". The supporting data is missing in the manuscript. The authors need to provide the details for this observation In Table 1.

Line no. 146, please correct the spelling of benzodiazepine and please check and correct the spelling mistakes, if any, throughout the manuscript.

In line no. 165, the authors need to elaborate on the meaning and significance of "the healthy survival effect" which was used several times in this manuscript.

In line 189, mention the expansion of ABDA.

Lines 196-197, "as the frequency of DDIs tended to decrease, and the total number of DDIs decreased". The authors need to provide the difference between "frequency" and the "total number" in this context.

Line nos 208-209, "UpToDate Classification was analyzed as a 2-drug combination, contrary to the Beers 2019 Criteria, which was analyzed as a >3-drug combination for the detection of DDI". The authors need to elaborate on the meaning of this sentence, for the benefit of the readers.

In line 203, mention the expansion of ACB.

Lines nos. 222-223, "The interviewed sample may not be represent of all the oldest-old people living in Turkey because the data were collected by the snowball technique". The authors need to elaborate on why they consider this as a limitation.

Lines nos. 223-224, states that "The study was conducted between December 2021 and May 2022, and the medications used by participants might be affected by seasonal effects". How did the authors get approval in April 2022, for a study that already started in December 2021? (as mentioned in line no. 82).

Line nos. 224-225, states that "The Beers 2019 Criteria and UpToDate Classification were not specifically developed for Turkey". I guess the DDI classification can not be country or region-specific. The authors need to explain why they felt/considered it as a limitation of this study.

The study conclusion does not match with the study objective. The objective was to assess the types of DDIs among the oldest-old population. Whereas the conclusion is related to the use of resources for checking DDIs. The authors are requested to align the conclusion as per the study objective, and results.

References nos. 3 & 7 need to be mentioned in English. Reference no. 17 must be as per the journal referencing style. Also please check and correct the reference style as per the journals' requirement.

Additional comments

The manuscript requires professional English editing. The authors may take help from a colleague who is a native English speaker.

The discussion, conclusion, and abstract need to be updated after attending to the suggestions/comments made to different sections of the manuscript (Please go through the reviewer comments mentioned above).

Reviewer 2 ·

Basic reporting

Title & Abstract
Title:
The abstract and title capture the content of the study well.
Abstract:
The study’s focus on drug-drug interactions (DDIs) within oldest-old age group in Turkey is timely and original, as the polypharmacy heightens the risk of DDIs leading to serious outcomes, including hospitalization, diminished quality of life, and even mortality. By examining DDI risks using the Beers Criteria and Lexicomp, it brings attention to inconsistencies in DDI classification between widely used guides, highlighting potential areas for improving the clinical decision making.

Introduction
The background and information provided adequate for understanding the research. However, authors may describe the oldest-old group of people in detail and mention other studies. In line 49 authors give one citation for the oldest old group in Turkey but may mention how this compares with other oldest-old people in other countries. The language of the manuscript is clear and professional English is used however, some sentences are lengthy. Simplifying sentence structure and using more direct language would improve flow. Authors also may describe this further by mentioning this reference study that describes the clinical features of the oldest old:
• Escourrou, E., Durrieu, F., Chicoulaa, B. et al. Cognitive, functional, physical, and nutritional status of the oldest old encountered in primary care: a systematic review. BMC Fam Pract 21, 58 (2020).
• Also, incorporating more recent research on polypharmacy in geriatrics, especially in the oldest-old, would reflect current trends and emerging concerns in the field. Reviewer suggests addition of the following reference:
• Maher RL, Hanlon J, Hajjar ER. Clinical consequences of polypharmacy in elderly. Expert Opin Drug Saf. 2014 Jan;13(1):57-65.
Also, the 28th reference in the list line 327- needs to get reformatted. The language of the manuscript is clear and professional English is used however some sentences are lengthy. Simplifying sentence structure and using more direct language would improve flow.

Figures & Tables
The figures and tables are clear and legible and free from unnecessary modification.

Experimental design

Material and Methods
The study identifies DDI prevalence but does not address the clinical impact or severity of these interactions, which limits the practical implications for healthcare providers. The methods section of this manuscript provides a general outline but lacks specificity in areas crucial for replication and clarity:
Lines 87-90: The authors describe using the "snowball sampling method" across Turkey to reach participants aged 85+. However, this method may introduce sampling bias, as it relies on participants' social networks and may not fully represent the population's diversity. A more randomized sampling approach could reduce bias.
• Lines 97-100: The study used the Beers 2019 Criteria and the UpToDate® Lexicomp® drug interaction guides. While these are established tools, the authors do not explain why these particular criteria were chosen over others. Including a rationale would strengthen the methodology.
• Line 102: The classification difference between the Beers Criteria (focused on 3-drug combinations) and UpToDate (2-drug combinations) is briefly mentioned but should be expanded. This distinction is significant, as it affects DDI identification and risk categorization. Providing more context on how this affects the study’s outcome would be valuable.
• Lines 104-110: The authors employed various non-parametric tests and correlation analyses to assess DDIs. However, there is limited justification for the choice of statistical methods, especially regarding whether they meet the assumptions required for these tests. Including information on why non-parametric tests were selected would improve clarity and robustness.
• Lines 109-110: The kappa test was used to calculate concordance between the two DDI criteria. Given that this is a primary aspect of the study’s methodology, additional details on the calculation and interpretation of kappa values would be beneficial.

Validity of the findings

Results
• The results are novel and the report on an understudied age group with high polypharmacy risks adds valuable data to geriatric pharmacology. However, the reviewer suggests some improvement: the uniqueness of the study should be emphasized by comparing these findings with previous research conducted in other countries or settings, which would highlight how the results contribute specifically to the Turkish context of oldest-old patients as that has already been suggested in the introduction part too.
• The mention of a “healthy survival effect” in the oldest-old group could be elaborated with potential mechanisms or hypotheses explaining why advanced oldest-old individuals might experience fewer DDIs. The results identify common DDI patterns, offering valuable information on which drug classes are most involved, which can guide healthcare providers in monitoring and managing prescriptions for the oldest-old. While the study shows DDI prevalence, it does not explore the clinical impact of these interactions. Adding these will add real-world relevance to the results. Grouping DDIs by severity (e.g., mild, moderate, severe) and discussing specific risks or consequences for each category could make the results more applicable for clinicians. Authors use Beers and Lexicomp but further explanation of these classifications and how they differ would help to understand the limitations of each system to improve DDI management strategies.

Discussion
The findings presented in the study align with the stated objectives and observed results, particularly regarding the high frequency of drug-drug interactions (DDIs) among the oldest-old population in Turkey. Key correlations are provided through data on the incidence of DDIs as associated with polypharmacy and medication types, as shown in Table 2 and Table 4, and the results are largely supported by statistical analysis that emphasizes the significant relationship between the number of medications and the prevalence of DDIs. Line 117-120: The study found that 54.6% of participants had at least one DDI, according to the UpToDate classification. This supports the hypothesis regarding the high DDI prevalence among the oldest-old population due to polypharmacy. Also, the frequency of DDIs decreases among the 100+ age group compared to younger cohorts, suggesting a healthy survival effect helps which helps to explain age-related trends and provides a broader understanding of the DDI risks. The findings are relevant to the study’s objectives, particularly in highlighting the impact of DDIs within an older demographic in Turkey, who are at high risk due to polypharmacy. This relevance is reinforced by comparing results with global studies (e.g., comparisons with data from Norway and Germany), which situates the study within a global context and demonstrates the broader implications for geriatric pharmacology. These global studies also show similar trends in drug-drug interactions (DDIs) among elderly populations in Norway and in Germany. And these global studies on oldest-old age group may be introduced already in the introduction section too.

Conclusion
The conclusions in this study align well with the findings obtained, particularly in emphasizing the frequency of drug-drug interactions (DDIs) among the oldest-old population in Turkey. The authors effectively highlight that using only one guide for DDI analysis may not comprehensively cover all types of interactions, advocating for the use of multiple guides to enhance the accuracy of DDI evaluations for this vulnerable age group. The findings align with the objectives set out in the study. By identifying discrepancies between the Beers Criteria and UpToDate Lexicomp guides and supporting their argument with global comparisons, the authors validate the need for multi-guide assessments to ensure patient safety, especially in settings with high polypharmacy risks.

Additional comments

No comments

---

## Round 0.2 · Minor Revisions

The author has made a significant effort to address the concerns of the reviewers. There are just a couple of smaller points that need attention.

1) The study was really about potential drug-drug interactions, as acknowledged by the authors. However, in their zeal to change DDI to pDDI, they have done so everywhere, including when talking about the work of others in the Introduction. The authors need to make sure that the references they previously used to support discussion of DDI (e.g. Lavan and Gallagher) are actually about potential and not actual DDI. I also note that there are no actual references to the studies described in lines 107-110, these need to be included.

2) It would be useful for the authors to describe why they can't know whether drugs were prescribed by a doctor or otherwise obtained by the participants. It makes sense to me when explained in the rebuttal letter, but readers of the article may well not know about electronic prescription workflow in Turkiye - if they did, it would help them understand why a potentially interesting analysis could not be done.

---

## Round 0.3 · accepted · Accept

Thank you for addressing the final small comments on the paper.